# Effect of a Multicomponent Intervention with Tele-Rehabilitation and the Vivifrail© Exercise Programme on Functional Capacity after Hip Fracture: Study Protocol for the ActiveFLS Randomized Controlled Trial

**DOI:** 10.3390/jcm13010097

**Published:** 2023-12-23

**Authors:** Bernardo Abel Cedeno-Veloz, Irache Casadamon-Munarriz, Alba Rodríguez-García, Lucia Lozano-Vicario, Fabricio Zambom-Ferraresi, María Gonzalo-Lázaro, Ángel María Hidalgo-Ovejero, Mikel Izquierdo, Nicolás Martínez-Velilla

**Affiliations:** 1Navarre University Hospital (HUN), Irunlarrea 3, 31008 Pamplona, Navarra, Spain; irache.casadamon.munarriz@navarra.es (I.C.-M.); alba.rodriguez.garcia@navarra.es (A.R.-G.); lucia.lozano.vicario@navarra.es (L.L.-V.); fabricio.zambom.ferrasi@navarra.es (F.Z.-F.); maria.gonzalo.lazaro@navarra.es (M.G.-L.); nicolas.martinez.velilla@navarra.es (N.M.-V.); 2Navarrabiomed, Institute for Health Research of Navarra (IDISNA), Irunlarrea 3, 31008 Pamplona, Navarra, Spain; mikel.izquierdo@gmail.com; 3Department of Health Sciences, Public University of Navarre, Av Cataluña s/n, 31006 Pamplona, Navarra, Spain; 4CIBER of Frailty and Healthy Aging (CIBERFES), Instituto de Salud Carlos III, Av Monforte de Lemos 3-5, Pabellón 11, Planta 0, 28029 Madrid, Spain; 5Department of Orthopaedics Clinics and Traumatology, University Hospital of Navarre (HUN), 31008 Pamplona, Navarra, Spain; angel.hidalgo.ovejero@navarra.es

**Keywords:** hip fracture, tele-rehabilitation, FLS, multicomponent intervention, physical exercise

## Abstract

Introduction: Hip fractures are the most common fracture leading to hospitalization and are associated with high costs, mortality rates and functional decline. Although several guidelines exist for preventing new fractures and promoting functional recovery, they tend to focus on osteoporosis treatment and do not take into account the complexity of frailty in older adults and geriatric syndromes, which are important factors in individuals at risk of suffering from frailty fractures. Moreover, most health systems are fragmented and are incapable of providing appropriate management for frail and vulnerable individuals who are at risk of experiencing fragility fractures. Multicomponent interventions and physical exercise using tele-rehabilitation could play a role in the management of hip fracture recovery. However, the effectiveness of exercise prescription and its combination with a comprehensive geriatric assessment (CGA) is still unclear. Methods: This randomized clinical trial will be conducted at the Hospital Universitario de Navarra (Pamplona, Spain). A total of 174 older adults who have suffered a hip fracture and fulfil the criteria for inclusion will be randomly allocated to either the intervention group or the control group. The intervention group will receive a multicomponent intervention consisting of individualized home-based exercise using the @ctive hip app for three months, followed by nine months of exercise using Vivifrail. Additionally, the intervention group will receive nutrition intervention, osteoporosis treatment, polypharmacy adjustment and evaluation of patient mood, cognitive impairment and fear of falling. The control group will receive standard outpatient care according to local guidelines. This research aims to evaluate the impact of the intervention on primary outcome measures, which include changes in functional status during the study period based on the Short Physical Performance Battery. Discussion: The findings of this study will offer valuable insights into the efficacy of a comprehensive approach that considers the complexity of frailty in older adults and geriatric syndromes, which are important factors in individuals at risk of suffering from frailty fractures. This study’s findings will contribute to the creation of more effective strategies tailored to the requirements of these at-risk groups.

## 1. Background and Rationale

Osteoporosis is a prevalent disease globally, and fragility fractures, especially hip fractures in older adults, impose a significant burden on health and economics [1,2]. Despite efforts to curb the increasing incidence of hip fractures, it remains a “silent epidemic” [1] affecting populations worldwide. The projected rise in the number of fragility fractures is alarming, and many fracture liaison services (FLS) primarily focus on bone metabolism treatments, therapeutic adherence and mortality [3], ignoring other critical factors that affect older adults. Among these factors, we find functional decline, cognitive impairment, malnutrition, frailty, sarcopenia, pain, falls and comorbidities [4].

FLS have not yet studied the special approach required for frail and vulnerable individuals at risk of experiencing fragility fractures [5,6,7]. Although there is a consensus on the importance of nutrition, calcium, vitamin D and certain osteoporosis medications [8], the effectiveness and suitability of exercise guidelines for older adults remain controversial [9].

Current best practices for post-hip fracture recovery, particularly regarding long-term functional recovery [10], are not well established, primarily due to the diversity of research methods and varying results among published studies [9]. Despite these differences, a multidisciplinary approach, including progressive resistance exercises and balance training, is strongly recommended. Early ambulation, weight-bearing exercises, training in activities of daily living, community-level rehabilitation, comorbidity management/complication prevention and nutritional support are also suggested [11]. These interventions are based on multiple factors related to functional recovery after a hip fracture, including biological characteristics such as age, sex, comorbidities, cognitive status, nutritional state and biochemical parameters [12]. Nowadays, social support is also significantly associated with this recovery [13], so tele-rehabilitation may provide essential support in this context [14].

Tele-rehabilitation is a new way of providing rehabilitation remotely through information and communication technologies [15]. The @ctivehip [16] application is an example of a programme that has shown promising results in enhancing functional recovery, physical autonomy, emotional status, concerns about falls and quality of life, as well as reducing the emotional state and perceived burden of informal caregivers [17]. However, these finding are based on a 12-week programme, and the long-term effectiveness of such programmes among older hip fracture patients, including exercise interventions such as Vivifrail and ActiveHip, and their combination with comprehensive geriatric assessment (CGA) remain uncertain, as most studies have focused on evaluating their short-term effects over a three-month intervention period.

This research aims at advancing clinical practice guidelines for promoting functional recuperation following a hip fracture with tele-rehabilitation (physical exercise based on the @ctivehip and Vivifrail programmes [18]), nutrition, secondary prevention of osteoporosis, polypharmacy adjustment and other major comorbidities. Pathways for clinical management for older adults who are at risk of chronic illnesses, moreover, than osteoporosis are essential to approach the complexity of these patients.

## 2. Objectives

### Hypothesis

We hypothesize that a multicomponent intervention with tele-rehabilitation and the Vivifrail exercise programme will improve hip fracture recovery at the 12-month follow-up.

## 3. Methods and Analysis

### 3.1. Trial Design

This study will follow the recommendations of the International Conference on Frailty and Sarcopenia Research ICFSR Task Force 2020 [19]. This is a prospective, randomized controlled trial (RCT), two-group repeated measures experimental design. Patients will be assigned in an equal distribution ratio of 1 to 1. It was registered in the Clinical Trials Registry at clinicaltrials.gov NCT05435534 (Date of registration 25 May 2022).

### 3.2. Study Setting 

The study will take place in the Department of Orthopaedics Clinics and Traumatology of Navarre University Hospital (Pamplona, Spain). Hospitalized patients who are eligible based on the screening criteria will be informed about the study’s details. After signing the consent form (Appendix A), participants will be allocated randomly to either the intervention group or the active control care group.

### 3.3. Eligibility Criteria and Recruitment 

The study participants will be older inpatient adults ≥ 75 years in the trauma ward of Navarre University Hospital (Pamplona, Spain) after a hip fracture. The Navarre University Hospital Research Ethics Committee granted approval for this research (PI_2022/7) on 25 April 2022. The trial commenced recruitment on 1 June 2022 and is currently open for recruitment.

Patients will be eligible to participate if the following apply: (i) age ≥ 75 years with a diagnosis of hip fracture fragility; (ii) Barthel index score for activities of daily living (ADL) of ≥60 (scale: 0, severe functional dependence; 100, functional independence) 2 weeks before fracture [20]; (iii) mobility independence on the Functional Ambulation Classification (FAC) scale of ≥3 (scale: 0, non-functional ambulatory; 5, independent ambulator) 2 weeks before fracture [21]; (iv) ability/support to use the ActiveHIP app (defined as the presence of a patient or caregiver willing to use the platform and ability to operate it after installing it on the cell phone in the presence of the recruiter and understand Spanish); and (v) obtaining informed consent from patients (if feasible), their family members or legal representatives.

Patients will be excluded if the following apply: (i) moderate–severe cognitive impairment with a Global Deterioration Scale (GDS) score of ≥5; (ii) secondary osteoporosis [22]; (iii) institutionalized in a permanent nursing home; (iv) patient’s refusal to provide informed consent, refusal by the primary caregiver or inability to obtain consent; (v) advanced illness with a life expectancy of less than 3 months; and (vi) any condition that prevents the engagement and/or safety in physical exercise, such as a recent heart attack within the past 3 months, unstable angina, severe heart valve insufficiency, uncontrolled high blood pressure or arrhythmia, recent pulmonary embolism within the past 3 months and hemodynamic instability. Only the conditions specifically mentioned will be taken into consideration.

### 3.4. Who Will Take Informed Consent?

Participants will be recruited through various methods, including posters and other informational materials. The consent process will involve providing explanations using an informed consent explanatory document and consent forms, and written consent will be obtained from all participants and their legal guardians. These consent forms will be under the scrutiny of the Ethics Committee to ensure all ethical standards will be met.

### 3.5. Additional Consent Provisions for Collection and Use of Participant Data and Biological Specimens 

The informed consent process allows for additional analyses of the gathered data. Blood samples will be collected.

### 3.6. Explanation of the Choice of Comparators 

#### 3.6.1. Interventions

In the active control group (control), participants will receive outpatient care in line with standard clinical practice. This sets it apart from traditional control groups in other studies that have no planned interventions. The intervention group (ActiveFLS), on the other hand, will receive an individualized multicomponent physical exercise programme based on the ActiveHip+ for 3 months, in addition to standard care. In subsequent revisions, after finishing the ActiveHip+ programme, the Vivifrail programme will be given according to the patient’s functional capacity. A CGA will be performed, evaluating nutrition status, polypharmacy, cognitive impairment and mood disorders. Nutritional intervention, adjustment of polypharmacy according to the Screening Tool of Older Person’s Prescriptions (STOPP) and Screening Tool to Alert to Right Treatment (START) criteria, and management of anxiety, depression, cognitive impairment and fear of falling will be conducted, as well as protocolized secondary fracture prevention treatment. Throughout the study, all participants will be permitted to maintain their regular physical activity levels. The interventions and follow-up are time-matched (Figure 1), ensuring both groups’ experiences are synchronized over the same time period.

#### 3.6.2. Intervention Description 

ActiveFLS intervention: We will propose a comprehensive geriatric assessment programme that includes a multicomponent physical exercise programme guideline based on ActiveHip+. ActiveHip+ is an asynchronous mobile app that is loaded onto the patient’s smartphone. Given the limitations of older adults with smartphone apps, the role of the caregiver will be essential in maintaining continuous oversight of the patient’s rehabilitation regimen.

The ActiveHip+ programme will feature a health education programme comprising five modules tailored for patients and caregivers, along with two extra modules exclusively for caregivers. These modules will provide information on hip fracture recovery and strategies to prevent a second fracture. A detailed description of the programme can be found in [23]. The home-based tele-rehabilitation programme, developed by a multidisciplinary team of health professionals and engineers, will include physical exercise and occupational therapy, with three smartphone-based sessions per week. The exercise programme will consist of two physical training sessions and one session of occupational therapy, ideally arranged on alternating days, with each session spanning 30 to 60 min.

We will provide exercise guidelines based on the Vivifrail programme. The focus of this programme is to provide personalized exercise plans consisting of multiple components, tailored to the functional abilities of older individuals and to be performed at home. The programme includes exercises for resistance/power, balance, flexibility and cardiovascular endurance. A detailed description of the Vivifrail programme can be found at http://vivifrail.com/resources/, accessed on 3 October 2023 [18].

After T3 (3-month assessment, Figure 1), participants in the intervention group will be assigned to one of the four tailored Vivifrail exercise programmes, determined by their physical functional level: Disability (scoring 0–3 on the SPPB), Frailty (4–6 points), Prefrailty (7–9 points), and Robus (10–12 points). Each participant will receive a duplicate of their individual exercise. 

The exercise intervention will consist of a 5-day-a-week routine of multicomponent exercises for 12 consecutive weeks. This routine will include exercises focusing on resistance, balance and flexibility three times a week, along with a walking schedule for five days a week. At the 6-month assessment, patients and caregivers will receive an updated exercise plan tailored to the patients’ functional condition at that point. This programme will remain the same until the final assessment.

A protocolized nutritional intervention will be carried out [11] based on the Global Leadership Initiative on Malnutrition (GLIM) criteria [24], with a focus on recommendations for protein intake, calcium and vitamin D [25]. Oral nutritional supplementation, if needed, will consist of supplements enriched in β-hydroxy-β-methylbutyrate (HMB) [26]. Vitamin D and anti-osteoporosis treatments will be prescribed following national guidelines [27], with zoledronic acid as the preferred choice due to better tolerance and adherence [28]. The patient’s treatment will be reviewed and adapted based on the STOPP/START criteria [29]. Additionally, the patient’s mood, cognitive impairment and fear of falling will be evaluated and addressed. The evaluation of depression will follow established clinical practices, utilizing a comprehensive approach that includes both pharmacological strategies, such as the use of prescribed medications, and non-pharmacological strategies, encompassing treatments such as psychotherapy, cognitive-behavioural therapy and lifestyle changes [30]. The training protocol is shown in Figure 1.

Active control care group (control): Individuals assigned to the standard care group will be provided with typical outpatient treatment. This consists of multidisciplinary and multicomponent follow-up during hospital admission by traumatology, rehabilitation and internal medicine/geriatrics. At discharge, a continuity of care report is made for follow-up by the primary care team as well as a 1-month review by traumatology with a control X-ray to check consolidation of the surgical fracture.

### 3.7. Participant Timeline 

The Barthel index, FAC scale, GDS and institutionalization status will be used as screening tests to assess the general functional capacity of the patient prior to the hip fracture. The study will have four major data collection points (baseline during acute hospitalization and at 3, 6 and 12 months) and one minor point (at 1 month). Table 1 displays the various times at which the different outcomes are measured. The study flow diagram is displayed in Figure 2.

### 3.8. Criteria for Discontinuing or Modifying Allocated Interventions 

Participants who are randomly selected for the intervention group will be motivated to utilize the Vivifrail programme and/or usual care completely and sequentially as prescribed. As this practice-level intervention poses a low risk, there are no predefined rules for early termination.

#### Strategies to Improve Adherence to Interventions 

This study will aim to promote intervention adherence by designing a multifactorial intervention rehabilitation programme after hip fracture based on a comprehensive geriatric assessment, secondary prevention of fracture and home-based rehabilitation with ActiveHip and Vivifrail intervention based on high-quality evidence of FLS follow-up and international guidelines. The assess adherence to the ActiveHip programme, the application performs weekly checks on the exercise sessions completed. If adherence falls below 50%, an automatic email is sent to the researcher responsible for patient follow-up. Adherence to the Vivifrail programme will be based on the patient’s daily record, which will be collected at each follow-up visit throughout the study.

### 3.9. Relevant Concomitant Care Permitted or Prohibited during the Trial 

During the trial’s follow-up, participants will not take part in other research projects that involve physical exercise interventions. However, participants are allowed to continue with any other non-conflicting interventions or therapies prescribed by their healthcare providers during the training period.

### 3.10. Provisions for Post-Trial Care 

Not applicable. The intervention in this study will be implemented as part of the usual clinical practice for 12 months. Participants will have access to post-trial care and may choose to incorporate other strategies to improve their medical practice through consultation with a physician.

## 4. Outcomes 

### 4.1. Primary Outcome

The main outcome to be measured is the change in functional status throughout the duration of the study. Functional status will be assessed using the Short Physical Performance Battery (SPPB) [31], a single tool that evaluates balance, gait ability and leg strength. The SPPB test has proven to be a reliable tool for evaluating functional capacity and quality of life following a hip fracture [32]. The overall score spans from 0, representing the lowest functional capacity, to 12 points, indicating the highest level of functional ability. A 1-point change in the score has been demonstrated to be clinical relevance [33].

### 4.2. Secondary Outcomes 

The secondary measures will assess constructs related to hip fracture, such as physical and cognitive decline, sarcopenia, nutrition, quality of life and healthcare system utilization. Furthermore, osteoporosis-related parameters will be measured using instrumented examinations, blood tests and dual-energy X-ray absorptiometry (DXA) (see Table 1).

−**Functional status:** The Barthel index of independence during ADL (0, worst; 100, best) [20], Lawton’s Instrumental Activities of Daily Living (IADL) scale (score from 0, worst, to 8, best) [34] and the FAC scale (0, non-functional ambulatory; 5, independent ambulator) [21] will be used.−**Cognitive status** [35]**:** The GDS, outlining seven distinct stages ranging from normal cognitive function to severe Alzheimer’s dementia, and the 16-item Informant Questionnaire on Cognitive Decline in the Elderly (IQCODE), with scoring for each question from 1 (significantly improved) to 5 (significantly worse), will be employed. An average score of 3.31/3.38 serves as the cut-off point, offering an equilibrium in detecting sensitivity and specificity of cognitive impairment [36]. Delirium assessment during hospitalization will be carried out with the Abbreviated Mental Test 4 (4AT) [37].−**Mood status:** Depression will be screened using the 15-item Yesavage Geriatric Depression Scale (scale: 0, best; 15, worst), which is independently associated with hip fracture [38]. Additionally, the level of fear regarding falls will be evaluated using the Falls Efficacy Scale International (FES-I), with validated thresholds for low concern (16–19 points), moderate concern (20–27 points) and high concern (28–64 points) [39].−**Frailty and sarcopenia:** The presence of frailty will be initially screened using the FRAIL questionnaire and further confirmed by the adapted criteria of Fried’s frailty [40]. Sarcopenia will be determined by: (i) handgrip strength < 16 kg for women or < 27 kg for men; and (ii) appendicular skeletal muscle mass (ASMM)/ height^2^ < 7.0 kg/m^2^ for men or < 5.5 kg/m^2^ for women [41]. Handgrip strength will be measured using the Groningen Elderly Test with a Smedley hand dynamometer [42]. We will record the best of three attempts (with a 30 s rest in between). Severe sarcopenia will be defined as gait speed ≤ 0.8 m/s or SPPB ≤ 8 points.−**Quality of life:** The EuroQol-5D and the Sarcopenia and Quality of Life (SarQoL) scales will be used to measure the quality of life: the former assesses five dimensions of health status and is a valid instrument for hip fracture patients [43], and the latter is a novel validated instrument for measuring the quality of life in sarcopenia patients [44].−**Other clinical assessments:** A comprehensive geriatric assessment will be conducted to evaluate geriatric syndromes [45], including falls (defined as an unplanned and involuntary loss of stability resulting in the individual unintentionally coming into contact with the ground), polypharmacy (defined as five or more medications) [46] and pain (Visual Analogue Scale: 0, best; 10, worst). Digital stadiometer will be used for height measured. The evaluation of nutrition will be conducted through the calculation of body mass index (BMI), determined by weight divided by height squared, and by administering the Mini-Nutritional Assessment (MNA) instrument [47]. Comorbidities will be evaluated with the Cumulative Illness Rating Scale for Geriatrics (CIRS-G) [48], ranging from 0 (best) to 56 (worst). Osteoporosis risk assessment is evaluated using the FRAX and QFracture tools [49], and pain is evaluated using the Visual Analogue Scale (VAS).−**Adverse events**: As per the International Conference on Harmonization Guidelines, a serious adverse event will be classified as any occurrence that leads to death, poses a threat to life, necessitates hospital admission or extends current hospitalization, causes lasting or substantial disability or is a congenital anomaly or birth defect [50].−**Use of health sources:** This will include hospital admissions, nursing home admissions, visits to primary care physicians and visits to the emergency department.−**Biochemical analyses:** Blood samples will be collected in Vacutainer tubes and centrifuged at 3300 rpm for 10 min at room temperature using a fixed-angle rotor. Following centrifugation, the serum from the top layer will be meticulously separated from the plasma in the lower layer, portioned into 100 μL aliquots and promptly preserved at −80 °C. Additionally, both plasma and buffy coat will be extracted and kept in polypropylene tubes at −80 °C until the time of analysis. Bone turnover markers (BTMs) will be measured at the Clinical Neuroproteomics Unit (Navarrabiomed), whereas other measurements will be performed at the Central Laboratory Unit of Navarra (LUNA). Biological samples will be collected following an overnight fast, between 8 and 10 am. Tests for 25-hydroxyvitamin D3 (vitamin D), calcium, phosphorus, alkaline phosphatase, parathyroid hormone (PTH), thyroid-stimulating hormone (TSH), creatinine and albumin will be conducted clinically right after the samples are delivered to the laboratory. Given the common occurrence of hypoalbuminemia in older adults, serum levels of albumin and calcium will be used to adjust the calcium value (corrected calcium value = Ca + 0.8 [40 − albumin]). This corrected calcium value will then be utilized in further analyses. Measurements of C-terminal cross-linked telopeptide of type I collagen (CTX), sclerostin (SCL), bone-specific alkaline phosphatase (B-ALP), procollagen type 1 N propeptide (P1NP) and osteocalcin (OC) will be carried out using enzyme-linked immunosorbent assays as per the manufacturer’s guidelines on the frozen samples [51].−**Dual-energy X-ray absorptiometry (DXA):** Bone mineral density (BMD) along with body composition, including fat and lean mass, will be evaluated using a Hologic DPX-IQ Discovery DXA device provided by GE Healthcare, located in Pollards Wood, UK. To minimize variability, all measurements will be performed by the same operator. The DXA machine will be calibrated daily. BMD will be gauged in grams per square centimetre at the non-dominant wrist, lumbar spine, and avaible proximal femur (encompassing the neck, trochanter, intertrochanteric area and Ward’s triangle) [52]. The L1 to L4 region will be included by positioning the patient in alignment with the table’s axis during examination. For BMD measurements in the proximal femur, the patient’s legs will be rotated 15–30° to subtly reveal the smaller trochanter of the femur. Z-scores and T-scores will be calculated at both sites, with a coefficient of variation set at 1.14%. Osteopenia and osteoporosis are defined according the World Health Organization standard criteria, which classify a BMD T-score between −1.0 SD and −2.49 SD below the young adult mean as osteopenia, and a score below −2.5 SD as osteoporosis [53]. Lean mass will be quantified as appendicular skeletal muscle mass (ASM), normalized either for height squared (resulting in the appendicular skeletal muscle mass index, ASMI) or for body mass index (ASM/BMI) [41].

### 4.3. Sample Size

Assuming an alpha error of α = 5%, the simple sample size that will be required to achieve a power of 90%, a ρ = 0.5, a standard deviation for the SPPB of σ = 2.5 and to detect a 10% difference in the frequency of patients obtaining a functional improvement of more than 1 point in the SPPB between each group will be 138 (69 per group), with an expected proportion of success in the usual clinical practice arm set at 30%. Given the characteristics of the study and the complexity of the patients (older adults after hip fracture) and taking into account an anticipated 20% dropout rate during the follow-up period, the calculated sample size is set at 174 participants, with 87 patients allocated to each arm of the study. These calculations are based on a two-sided test. The 10% difference between both the intervention and the control group, according to data from a previously published home-based rehabilitation intervention on hip fracture patients [10], representing a functional improvement greater than 1 point in the SPPB at 12 months between each group, will be considered clinically relevant based on the most relevant clinical variables involved in the functional decline after hip fracture [54,55].

### 4.4. Assignment of Interventions: Allocation

#### 4.4.1. Sequence Generation

Prior to randomization, the research staff will review the absolute and relative contraindications to participating in the exercise programme and will provide general information about the study. Eligible practices will be allocated to either the intervention or control group, utilizing a stratified randomization process. This will be facilitated using a computer-generated randomized block approach, with blocks of four, ensuring equal distribution across various characteristics (e.g., practice size, location). The randomization will be conducted by an independent statistician, using the website www.randomizer.org, accessed on 3 October 2023. This method aims to minimize selection bias and ensure comparability between groups right from the study outset.

#### 4.4.2. Concealment Mechanism

The allocation of participants to different groups is concealed from the research staff responsible for randomization, ensuring the integrity of the randomization process. To ensure masking, an alphanumerical code will be assigned to each study group. This code is delivered to the associated researcher and is not revealed to the investigators in charge of processing the data until the analysis of the coded interventions is completed. The allocation mechanism is carried out until 174 total patients are assigned. This is achieved by using a computer-generated randomization schedule managed by an independent investigator. While the nature of our intervention prevents blinding of the treatment to staff involved in training, the assessment staff at the clinic remain blinded to the participant’s group assignment, study design and expected outcomes. Patients and their families are informed about their random inclusion in a group without specifying which one. Upon request, this information can be disclosed to them.

#### 4.4.3. Implementation 

Upon determining a participant’s eligibility and readiness for randomization, one of the research staff will determine which block group they belong to and open the next randomization block. The principal investigator will be notified of the site’s randomization status and then will send an email to the practice and inform the study staff.

### 4.5. Assignment of Interventions: Blinding

#### 4.5.1. Who Will Be Blinded 

After a study participant is randomized, the specific study arm to which they are assigned will not be concealed. Nonetheless, the principal investigator, evaluators and staff responsible for data analysis will remain unaware of the identities of the participants within each intervention group.

#### 4.5.2. Procedure for Unblinding if Needed 

This statement is not relevant, as the study in question is an open intervention carried out at the level of practice.

### 4.6. Data Collection Methods (Plans for Assessment and Plans to Complete Follow-Up) and Data Outcome Management

At each visit, data collection and procedures will be carried out. The study data will be stored on an encrypted hard disk partition that can only be accessed by the research team. Only authorized researchers will have access to this password. Participants will be identified using numbers or symbols, and any information that could easily identify them (such as name or address) will not be stored in the dataset. In the event that a participant exits the study prematurely, they will be regarded as off-study and will adhere to the same schedule of events as those who remain in the study.

### 4.7. Confidentiality 

The study will adhere to the Spanish regulations, including Law 3/2018 (5 December 2018), for safeguarding personal information and ensuring digital rights; the European Union Parliament’s Regulation 2016/679 (dated 27 April 2016) pertaining to data protection (GDPR); and Law 41/2002 (14 November 2002), which is a basic regulatory law on patient autonomy.

## 5. Statistical Methods

### 5.1. Statistical Methods for Primary and Secondary Outcomes

We will incorporate all participants as originally allocated post-randomization in order to use the intention-to-treat approach. Missing data due to dropouts or deaths will be addressed using multiple imputations. We will calculate frequencies and confidence intervals in an initial descriptive analysis for qualitative variables. For continuous variables, we will report statistics of central tendency and dispersion, such as means, standard error and confidence intervals or the median and interquartile range. We will check the normality of continuous variables graphically and through K–M and Shapiro–Wilk tests, comparing their differences between groups using either parametric tests (*t*-tests, mixed-effects models) or non-parametric tests (Mann–Whitney U, Kruskal–Wallis). We will employ a Bonferroni post hoc test to evaluate statistically significant (*p* < 0.05) group and time differences. Spearman’s (rho) rank correlation coefficients and level of significance (*p*) will be used to assess the relationship between clinical/functional parameters and biochemical parameters, adjusted for age and sex. The values of *r* will be used to indicate small (*r* = 0.10), medium (*r* = 0.30) and large (*r* = 0.50) size correlations (i.e., effect size). Finally, we will assess the relationship between categorical and dichotomous variables through χ2 and Fisher exact tests. The level of statistical significance will be set at 0.05. We will analyse the data using SPSS package 23.0.

### 5.2. Interim Analyses 

Not applicable. The study will not include interim analyses or stopping guidelines since the medical practice-level intervention is considered low-risk.

### 5.3. Methods for Additional Analyses (e.g., Subgroup Analyses) 

An ancillary examination of the primary endpoint will incorporate variables identified prior to randomization that could potentially predict positive outcomes. These groups will include frailty, sarcopenia, osteosarcopenia, the degree of cognitive impairment and the overall adherence to the rehabilitation programme (low < 30%, medium 30–60% and high > 60%). A Bonferroni post hoc analysis will be employed to assess statistically meaningful differences (*p* < 0.05) between groups and over time.

### 5.4. Oversight and Monitoring

#### 5.4.1. Composition of the Coordinating Centre and Trial Steering Committee

Trial Steering Committee: Nicolás Martinez-Velilla (Chair), Robinson Ramírez-Vélez and María Gonzalo Lázaro.

Data Monitoring Committee: Mikel Izquierdo (Chair), Fabrizo Zambom-Ferrasi and Lucia Lozano-Vicario.

#### 5.4.2. Composition, Role and Reporting Structure of the Data Monitoring Committee

The ActiveFLS study will have an independent data and safety monitoring committee that advises the investigators. The committee members will provide their expertise and recommendations in an individual capacity and report directly to the principal investigator.

#### 5.4.3. Adverse Event Reporting and Harms

To ensure safety, the incidence of falls and serious injuries resulting from falls will be tracked. Information regarding falls will be derived from medical records throughout the follow-up period. Other adverse events relative to the intervention protocol (nutrition, vitamin D, osteoporosis treatment, etc.) will also be monitored. The research team will oversee data monitoring to record any minor or significant occurrences that might be related to either the intervention or the standard care groups throughout the study. The chief investigators will review any adverse events or unintended effects detected.

#### 5.4.4. Frequency and Plans for Auditing Trial Conduct

There will not be plans for auditing trial conduct.

#### 5.4.5. Plans for Communicating Important Protocol Amendments to Relevant Parties (e.g., Trial Participants, Ethical Committees)

Modifications to the study protocol will be digitally relayed to all research team members and will undergo examination in accordance with the guidelines of the Institutional Review Board.

#### 5.4.6. Dissemination Plans

Dissemination is a recurring item on the agenda for the Department of Orthopaedics Clinics and Geriatrics of Navarre University Hospital (Pamplona, Spain) and the International Conference on Frailty and Sarcopenia Research ICFSR Task Force 2020 [19]. Patient advisors will be involved in reviewing all study materials to ensure that the findings are presented in an understandable and usable way for a broad audience. The study results will be disseminated in various formats, including peer-reviewed publications, conference presentations, blog posts and policy briefs.

## 6. Discussion

For this study, we will be developing a multifactorial intervention rehabilitation programme after hip fracture. The programme will be based on a comprehensive geriatric assessment, secondary prevention of fractures and home-based rehabilitation with ActiveHip and Vivifrail. We will aim to examine whether this intervention could improve functional status after hip fracture. Our ActiveFLS intervention will be developed based on high-quality evidence of FLS follow-up [56,57] and international guidelines [11,27] on hip fracture management, and it is feasible for most types of patients with little support. The use of integrated models of care based on comprehensive geriatric assessment can help align clinical practice with the individual needs of patients and enhance their quality of life [58]. Due to the crucial role of supervision during exercise programmes on fracture reduction [59], this protocol will try to adapt current exercise programmes to produce consistent supervision and monitoring results.

One reason why home exercise programmes often fail is due to a lack of medium-term adherence [5,10,12]. The proposed exercise tools will assist in maintaining medium- to long-term adherence. For instance, a study with ActiveHip demonstrated that 63% of participants completed more than 20 sessions [17]. On the other hand, the Vivifrail programme showed 68% adherence at 3 months. While this figure may be debatable, it is higher than in other studies involving similar home-based exercise interventions [60]. With strategies to improve adherence to interventions, we aim to achieve similar or better adherence rates.

This study will have several strengths. First, it will be a combination of multiple interventions that were studied separately. This will also generate a problem in that the hypothetical expected benefit cannot be attributed to a specific intervention. However, given the complexity of managing older adults after a hip fracture, an approach in this direction will be possible to provide greater benefits. Secondly, adults aged 90 years and older will be included with a few exclusion criteria, making this study of broad impact on this heterogeneous population. Thirdly, it will be easily applicable to various regions as it is based on home-based rehabilitation and will not require any specific infrastructure for implementation. The study will also have several limitations. Firstly, it will not include patients with advanced dementia, defined as GDS ≥ 5 (a group with a high incidence of hip fracture), because the exercise interventions will not be adapted to this type of population [61]. Secondly, secondary osteoporosis will be also an exclusion criterion due to the variability of management in this population [22]. Thirdly, nursing home patients will be excluded from the study due to the difficulty of follow-up and adherence to the intervention protocol (especially tele-rehabilitation). It should be noted that the usual care group, although involved in the study, will receive certain components of the ActiveFLS intervention. This is because this arm will include an assessment by internal medicine/geriatrics and a follow-up by primary care. 

To our knowledge, many studies have been developed for hip fracture management, but they usually address issues from the fracture separately (exercise [59], nutrition [62], osteoporosis management [3]) or have low-quality evidence. If our hypothesis will be confirmed and demonstrate that our multifactorial and multicomponent programme will improve functional status, it will lead to the development of a new targeted therapeutic pathway for use after hip fracture discharge.

### 6.1. Contribution to the Field

Hip fractures are a common complication associated with osteoporosis and are known to lead to heightened morbidity, mortality and poorer functional recovery. Despite the numerous studies carried out in recent years, the best management in complex cases is still lacking. We hypothesize that multicomponent intervention with tele-rehabilitation could have a role in the evolution of hip fracture, given its multiple levels. This research represents the initial assessment of the effect of a multifactorial intervention that includes tele-rehabilitation based on physical exercise on the recovery of hip fracture patients. If our findings align with our expectations, a possible new pathway and therapeutic protocol after hip fracture could be developed and implemented.

### 6.2. Trial Status

The trial began enrolling participants on June 1, 2022, and is presently accepting new recruits. The recruitment process will conclude once 174 participants have been randomized. It is expected that this goal will be achieved by December 2025.

## 7. Ethics and Dissemination

### 7.1. Ethics Statement 

This study was approved by the Navarre University Hospital Research Ethics Committee (PI_2022/7) on 25 April 2022. At the point of screening and enrolment, we will acquire written consent from participants or their legal representatives using two distinct documents. To guarantee participant understanding, we will employ meticulous and comprehensive explanations while securing consent during both the screening and enrolment processes.

### 7.2. Availability of Data and Materials

Within 12 months of the conclusion of the study, we will publicly release the de-identified participant-level data used for analysing research questions through an online data repository.

## Figures and Tables

**Figure 1 jcm-13-00097-f001:**
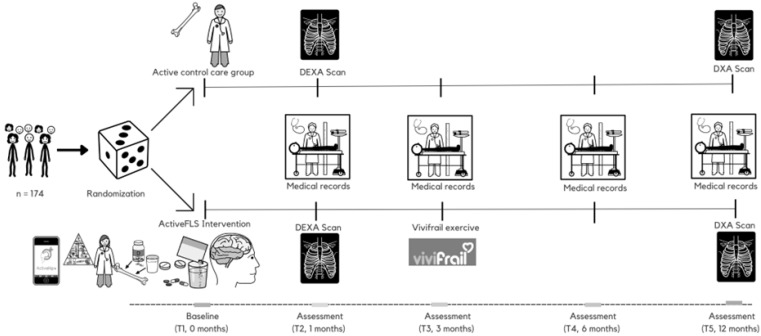
Intervention timeline through the “ActiveFLS randomized control trial”. Participants will be randomly assigned to intervention group (ActiveFLS Intervention, n = 87) or control group (active control care, n = 87). T: time point; DXA: dual energy X-ray absorptiometry.

**Figure 2 jcm-13-00097-f002:**
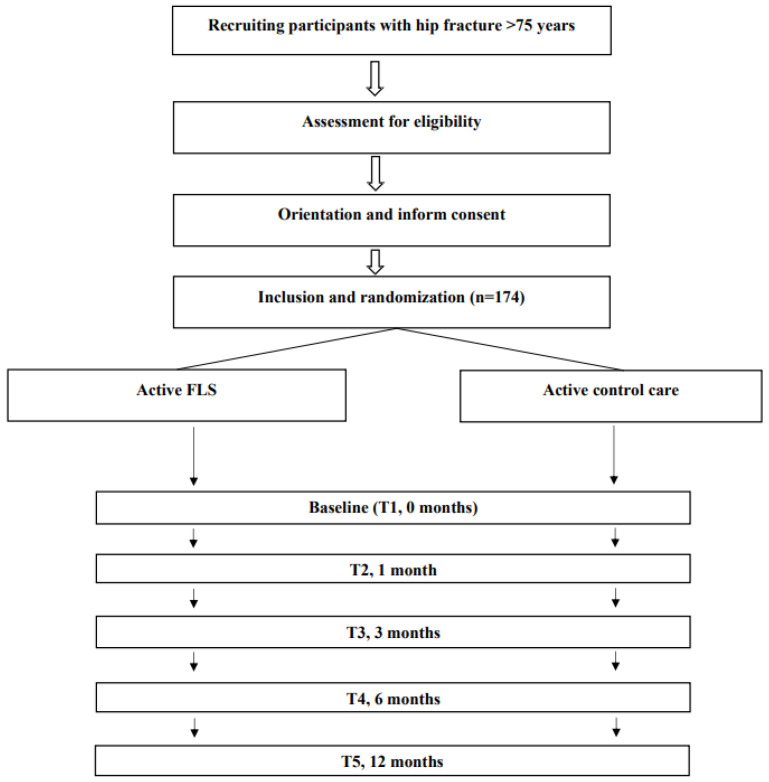
Flow diagram of the study protocol.

**Table 1 jcm-13-00097-t001:** Schedule for the different primary and secondary variables for the participants of the study.

Measure	Screening	T1 Baseline	T2 1 Month	T3 3 Months	T4 6 Months	T5 12 Months
Primary outcome						
Short Physical Performance Battery (SPPB)		x	x	x	x	x
Secondary outcomes						
Barthel index	x		x	x	x	x
Functional Ambulation Classification (FAC)	x		x	x	x	x
Lawton’s Instrumental Activities of Daily Living (IADL)		x	x	x	x	x
Global Deterioration Scale (GDS)	x		x	x	x	x
Mini-Mental State Examination (MMSE)		x	x	x	x	x
Abbreviated Mental Test 4 (4AT)		x	x	x	x	x
Yesavage Geriatric Depression Scale		x	x	x	x	x
Falls Efficacy Scale International (FES-I)		x		x		x
Frailty		x	x	x	x	x
Handgrip		x	x	x	x	x
Quality of Life (EuroQol-5D)		x	x	x	x	x
Sarcopenia and Quality of Life (SarQoL)		x		x		x
FRAX, QFracture		x				x
Urinary incontinence		x	x	x	x	x
Fecal incontinence		x	x	x	x	x
Pressure ulcers		x	x	x	x	x
Constipation		x	x	x	x	x
Polypharmacy		x	x	x	x	x
Rate and risk of falls		x	x	x	x	x
Visual Analogue Scale for Pain (VAS)		x	x	x	x	x
Cumulative Illness Rating Scale for Geriatrics (CIRS-G)		x				
Mini-Nutritional Assessment (MNA)		x	x	x	x	x
Adverse effects			x	x	x	x
Mortality			x	x	x	x
Admission and readmission to the hospital			x	x	x	x
Institutionalization	x		x	x	x	x
Blood test		x	x	x	x	x
Bone turnover markers (BTMs)			x			x
Dual-energy X-ray absorptiometry (DXA)			x			x

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
