# Peer review of "Effect of a Multicomponent Intervention with Tele-Rehabilitation and the Vivifrail© Exercise Programme on Functional Capacity after Hip Fracture: Study Protocol for the ActiveFLS Randomized Controlled Trial"

_jcm, 2023, doi:10.3390/jcm13010097_

Round 1
Reviewer 1 Report
Comments and Suggestions for Authors
Dear authors,
congratulations for your idea and procedures to try to make this project a reality. Multicomponent intervention with tele-rehabilitation is not new but there are a lot of things that are not explored. Specific to your topic I think this is a common problem in this type of population that the literature should debate more to tri to discover more strategies to improve the health of the older.
I think the program is well designed but i have some worries about the control of the intervention program. Who you will guarantee that the program is running like you programmed? I think that should be one or two control points to ensure the running of the program in the right way. This is my only worries that other than this i will be happy to read the results in a near future.
Congratulations
Author Response
Dear Reviewer,
Thank you for your valuable feedback on our project involving multicomponent intervention with tele-rehabilitation. Regarding your concern about monitoring the intervention program, we have robust measures in place.
For the ActiveHip program, the application performs weekly checks on the exercise sessions completed. If adherence falls below 50%, an automatic email is sent to the researcher responsible for patient follow-up.
The adherence to the Vivifrail programme will be based on the patient's daily record, which will be collected at each follow-up visit throughout the study, as it is included in the actual manuscript.
We have just modified the “Strategies to improve adherence to interventions” part in order to include this information
We are committed to maintaining the highest standards in our study and look forward to sharing our findings.
Sincerely,
The Authors
Reviewer 2 Report
Comments and Suggestions for Authors
This RCT study addresses a highly significant area of clinical research - i.e., recovery after osteoporosis-related hip fracture – and has the potential to contribute meaningfully to the field. Investigators should be congratulated for leading such an important effort. The report is well drafted, and the study design carefully thought out. The current version of the draft would benefit from minor revisions primarily for clarification of a few methodological issues, and inclusion of additional information.
1- Background and Rationale. This section could be substantially strengthened if it were more focused on the key problem addressed by the proposed intervention; i.e., recovery post hip fracture. The initial part of the Background addresses osteoporosis and the importance of prevention of osteoporosis-related hip fracture, but brings little background information on what is known about recovery. This section would benefit from more robust background information on what constitutes current best clinical practice guidelines post hip fracture, details about the epidemiology of functional recovery changes post hip fracture (regarding the latter, the Baltimore Hip Studies may be a useful source for relevant epidemiologic insight), and further insight into current experience in the filed regarding telehealth for rehabilitation, even if findings to date have been primarily on short -term outcomes.
2- Concealed allocation (section 4.4.2, line 346). The text is confusing, as it seems to mix up the two inter-related but different processes of concealed allocation prior to randomization (to enhance integrity of randomization process) and masking of treatment post randomization. In its current format, this section addresses only masking/blinding. Even if it is not possible to mask the treatment to which a participant was assigned, it is still possible to implement procedures to optimize allocation concealment from the research staff in charge of randomization.
3- Study period. Text says that “it is estimated that the study dates will be from 1 June 2022 …”. Has the study already started? Text needs to be revised for accurate information on study period and verb tense.
4- Arrhythmia is listed as an exclusion criterion. Will patients with well-controlled, chronic atrial fibrillation be eligible or any type of arrhythmia, whether controlled or not, will be excluded?
5- Information on the randomization process is very limited. Recommend revision to improve description of the randomization process.
6- The tele-rehabilitation aspect of the intervention is an important component. Readers will likely have a question regarding adherence to the telehealth program. In this context, it would be useful if investigators could expand a bit on positive experiences of Vivifrail and Active Hip, and elaborate on potential barriers/challenges regarding adherence to those tele-rehabilitation programs.
7- It would be useful to expand on the method for assessment of adherence to the telehealth intervention program. Text (lines 222-223) informs that adherence will be based on “daily records” collected at each follow-up visit throughout study. Will there be an index of adherence derived directly from electronic records or all will be based on daily diaries registered by participants and/or their caregivers over 1 year period?
8- Please clarify if study intervention will have both synchronous and asynchronous components.
9- Sample size calculation. The assumptions regarding % improving >=1 point in the SPPB in 12 months in the usual care and the effect size, were they based on preliminary data or literature review?
10- Table 1. It should be indicated that the VAS is regarding the assessment of pain. It would be useful to list the geriatric syndromes that will be assessed (frailty, sarcopenia, mobility disability, falls, other?)
11- Abstract, line 1: Is hip fracture the most common reason for overall hospital admission? If this is incorrect, as suspected, the sentence needs to be revised for accuracy.
12- Line 461. The term “…very old …” is not optimal, given that it could be seen as having a negative connotation. Consider indicating the age range that is often excluded from studies but that will be included in this study.
13- The use of the similar acronyms GDS and YE-GDS is confusing. For example, authors could use GDS for the Geriatric Depression Scale, and spell out the Global Deterioration Scale.
14- Line 203. I believe there is a typo, as it seems that the screening test will assess the patient’s capacity prior to hip fracture, and not capacity at a previous hip fracture.
15- Line 430. A verb is missing.
Author Response
This RCT study addresses a highly significant area of clinical research - i.e., recovery after osteoporosis-related hip fracture – and has the potential to contribute meaningfully to the field. Investigators should be congratulated for leading such an important effort. The report is well drafted, and the study design carefully thought out. The current version of the draft would benefit from minor revisions primarily for clarification of a few methodological issues, and inclusion of additional information.
1- Background and Rationale. This section could be substantially strengthened if it were more focused on the key problem addressed by the proposed intervention; i.e., recovery post hip fracture. The initial part of the Background addresses osteoporosis and the importance of prevention of osteoporosis-related hip fracture, but brings little background information on what is known about recovery. This section would benefit from more robust background information on what constitutes current best clinical practice guidelines post hip fracture, details about the epidemiology of functional recovery changes post hip fracture (regarding the latter, the Baltimore Hip Studies may be a useful source for relevant epidemiologic insight), and further insight into current experience in the filed regarding telehealth for rehabilitation, even if findings to date have been primarily on short -term outcomes.
Thank you for your insightful comments. We will revise this section to provide a more comprehensive understanding of these aspects, particularly emphasizing long-term recovery processes. Our aim is to incorporate findings from the Baltimore Hip Studies for epidemiological insights and to elaborate on the role of tele-rehabilitation, considering both short-term and potential long-term outcomes (line 74-83). This enhancement will align the background more closely with the study's objectives and contribute to a clearer understanding of the field's current state.
2- Concealed allocation (section 4.4.2, line 346). The text is confusing, as it seems to mix up the two inter-related but different processes of concealed allocation prior to randomization (to enhance integrity of randomization process) and masking of treatment post randomization. In its current format, this section addresses only masking/blinding. Even if it is not possible to mask the treatment to which a participant was assigned, it is still possible to implement procedures to optimize allocation concealment from the research staff in charge of randomization.
Thank you for your insightful comments regarding the concealed allocation section of our manuscript. We appreciate your feedback on the confusion caused by the current text, which focuses more on masking/blinding than on the process of concealed allocation prior to randomization. To address your concerns, we will revise this section to clearly differentiate between concealed allocation and post-randomization treatment masking
3- Study period. Text says that “it is estimated that the study dates will be from 1 June 2022 …”. Has the study already started? Text needs to be revised for accurate information on study period and verb tense.
The trial commenced recruitment on 1 June 2022 and is currently open for recruitment. Recruitment will cease when 174 participants have been randomized. It is anticipated that this target will be reached by December 2025. We have just modify this apart in order to clarify this information (see also Trial Status at the end of Discussion section)
4- Arrhythmia is listed as an exclusion criterion. Will patients with well-controlled, chronic atrial fibrillation be eligible or any type of arrhythmia, whether controlled or not, will be excluded?
Apologies for the error in writing. We meant to refer to 'uncontrolled arrhythmia'. Section VI of the exclusion criteria is based on safety considerations to prevent adverse effects related to exercise.
5- Information on the randomization process is very limited. Recommend revision to improve description of the randomization process.
We have added the 'Prior Randomization' information and expend the information about https://www.randomizer.org/. This section has been improved to enhance the concealment mechanism and provide detailed information about this process
6- The tele-rehabilitation aspect of the intervention is an important component. Readers will likely have a question regarding adherence to the telehealth program. In this context, it would be useful if investigators could expand a bit on positive experiences of Vivifrail and Active Hip, and elaborate on potential barriers/challenges regarding adherence to those tele-rehabilitation programs.
Thank you for your insightful comments. We appreciate your suggestion to expand on the experiences and challenges related to adherence in programs such as Vivifrail and Active Hip.In response, we have added a new section in the discussion part of our manuscript. This section addresses the adherence challenges often faced in home exercise programs and outlines how our proposed tools, ActiveHip and Vivifrail, have demonstrated relatively high adherence rates in past studies. This addition aims to provide a comprehensive understanding of the effectiveness and challenges in tele-rehabilitation adherence, enhancing the manuscript's value for readers interested in this aspect of the intervention.
7- It would be useful to expand on the method for assessment of adherence to the telehealth intervention program. Text (lines 222-223) informs that adherence will be based on “daily records” collected at each follow-up visit throughout study. Will there be an index of adherence derived directly from electronic records or all will be based on daily diaries registered by participants and/or their caregivers over 1 year period?
Thank you for your valuable feedback on our project involving multicomponent intervention with tele-rehabilitation. For the ActiveHip program, the application performs weekly checks on the exercise sessions completed. If adherence falls below 50%, an automatic email is sent to the researcher responsible for patient follow-up.We have just modified the “Strategies to improve adherence to interventions” part in order to include this information
In response to your inquiry about adherence indices in our study, we plan to utilize both electronic records and daily diaries maintained by participants and their caregivers. Our main analysis will adopt an intention-to-treat approach, considering all participants regardless of their adherence level. Additionally, we've included in our secondary analysis an assessment of overall adherence to the rehabilitation program, categorized as low (<30%), medium (30-60%), and high (>60%)
8- Please clarify if study intervention will have both synchronous and asynchronous components.
ActiveHip is an asynchronous telehealth app, facilitating non-real-time communication between healthcare providers and patients. This feature allows users to access the platform at their convenience, offering flexibility in terms of time and location. We have incorporated this characteristic into the description of the intervention for clarity
9- Sample size calculation. The assumptions regarding % improving >=1 point in the SPPB in 12 months in the usual care and the effect size, were they based on preliminary data or literature review?
This assumption is based on data from a previously published home-based rehabilitation intervention on hip fracture patients. We add the information to the Sample size part of the manuscript.
10- Table 1. It should be indicated that the VAS is regarding the assessment of pain. It would be useful to list the geriatric syndromes that will be assessed (frailty, sarcopenia, mobility disability, falls, other?)
We have modifed the table 1 to indicate that VAS is for pain assessment.
The only geriatric syndromes evaluated in the study that are not already included in Table 1 are urinary incontinence, fecal incontinence, pressure ulcers, and constipation. I am modifying Table 1 to include this information.
11- Abstract, line 1: Is hip fracture the most common reason for overall hospital admission? If this is incorrect, as suspected, the sentence needs to be revised for accuracy.
Apologies for the confusion in our earlier statement. We meant to emphasize that hip fractures are the most common fracture that the leading causes of hospitalization in older adults. We have revised this in the abstract. Thank you for pointing out the need for this correction.
12- Line 461. The term “…very old …” is not optimal, given that it could be seen as having a negative connotation. Consider indicating the age range that is often excluded from studies but that will be included in this study.
We appreciate your suggestion and agree that specifying the age range would be more appropriate and clearer. We have referred to individuals aged 90 years and older, a demographic often excluded from similar research. Accordingly, we have revised the manuscript to reflect this.
13- The use of the similar acronyms GDS and YE-GDS is confusing. For example, authors could use GDS for the Geriatric Depression Scale, and spell out the Global Deterioration Scale.
Thank you for pointing out the confusion caused by the use of similar acronyms, GDS and YE-GDS, in our manuscript. In the revised manuscript, we will use GDS to denote the Global Deterioration Scale and will spell out the Geriatric Depression Scale in full to avoid any confusion.
14- Line 203. I believe there is a typo, as it seems that the screening test will assess the patient’s capacity prior to hip fracture, and not capacity at a previous hip fracture.
Thank you for pointing out the typographical error. We have corrected it as per your suggestion.
15- Line 430. A verb is missing.
Sorry for the miss-verb. Phrase have changed to “There will not be plans for auditing trial conduct”
Thank you once again for your invaluable input. Your meticulous review and suggestions have significantly improved the clarity and comprehension of our protocol. We deeply appreciate your contribution to enhancing the quality of our work.
